# Monitoring interdecadal coastal change along dissipative beaches via satellite imagery at regional scale‡

Marcan Graffin[1,2] ⓘ, Mohsen Taherkhani[3] ⓘ, Meredith Leung[4], Sean Vitousek[5,6] ⓘ, George Kaminsky[7] and Peter Ruggiero[4] ⓘ

[1]LEGOS (CNES/CNRS/IRD/UT3), Université de Toulouse, Toulouse, France; [2]Lab'OT (CNES), Toulouse, France; [3]Department of Civil and Construction Engineering, Oregon State University, Corvallis, OR, USA; [4]College of Earth, Ocean, and Atmospheric Sciences, Oregon State University, Corvallis, OR, USA; [5]Pacific Coastal and Marine Science Center, U.S. Geological Survey, Santa Cruz, CA, USA; [6]Department of Civil, Materials, and Environmental Engineering, University of Illinois Chicago, Chicago, IL, USA and [7]Washington State Department of Ecology, Olympia, WA, USA

## Research Article

**Keywords:**
shoreline; waterline; remote sensing; coastal change

**Corresponding author:**
Marcan Graffin;
Email: marcan.graffin@ird.fr

‡This article has been updated since its original publication. A notice detailing these changes can be found here: https://doi.org/10.1017/cft.2024.3

## Abstract

Coastal morphological changes can be assessed using shoreline position observations from space. However, satellite-derived waterline (SDW) and shoreline (SDS; SDW corrected for hydrodynamic contributions and outliers) detection methods are subject to several sources of uncertainty and inaccuracy. We extracted high-spatiotemporal-resolution (~50 m-monthly) time series of mean high water shoreline position along the Columbia River Littoral Cell (CRLC), located on the US Pacific Northwest coast, from Landsat missions (1984–2021). We examined the accuracy of the SDS time series along the mesotidal, mildly sloping, high-energy wave climate and dissipative beaches of the CRLC by validating them against 20 years of quarterly *in situ* beach elevation profiles. We found that the accuracy of the SDS time series heavily depends on the capability to identify and remove outliers and correct the biases stemming from tides and wave runup. However, we show that only correcting the SDW data for outliers is sufficient to accurately measure shoreline change trends along the CRLC. Ultimately, the SDS change trends show strong agreement with *in situ* data, facilitating the spatiotemporal analysis of coastal change and highlighting an overall accretion signal along the CRLC during the past four decades.

## Impact statement

Coastal environments, particularly sandy beaches, are constantly changing on various temporal and spatial scales. Therefore, monitoring coastal change over different spatiotemporal scales is paramount for coastal scientists, managers and policymakers. Shoreline positions are a commonly used metric for evaluating coastal change. Historically, shoreline position data have been relatively scarce, except for a few locations of specific research interest worldwide, primarily due to the cost and labor required to collect data. In recent years, and thanks to newly developed techniques and models, shoreline positions extracted from satellite imagery have provided high-spatiotemporal-resolution data sets of coastal evolution. However, these data sets can often be subject to biases and uncertainties, which limit their applications, especially at high-energy sites. As a case study, we assess the accuracy of satellite-derived shoreline positions by comparing them to field observations of shoreline positions along the mesotidal, high wave-energy and dissipative sandy beaches of the Columbia River Littoral Cell (CRLC) in the US Pacific Northwest. Our findings indicate that after removing outliers and correcting the satellite-derived waterline data for tides and wave runup, a strong agreement is detected between the satellite-derived and field-observed shoreline positions along the CRLC, while removing outliers alone is sufficient to extract accurate shoreline change trends. These findings underscore that the transition from data scarcity to data abundance for shoreline positions, made possible by advancements in satellite remote-sensing techniques, can drastically enhance coastal monitoring throughout the world, particularly for regions that have been historically data-poor. These rich data sets can be employed for monitoring coastal change with high spatiotemporal resolution and will inform coastal communities, policymakers and planners regarding historical trends and patterns, assisting them in devising plans for the prevention and adaptation to potential future coastal hazards.

## Introduction

Coastal regions support economic and recreational activities as well as rich ecosystems. Yet these regions are constantly changing since they are subject to a myriad of hydrodynamic and geomorphic processes across temporal scales ranging from seconds to millennia and from

submeter to global scales. The dynamic evolution of coasts is likely to have accelerated in many parts of the world over the past few decades, notably due to anthropogenic drivers (Syvitski et al., 2022). Although there is no consensus on the long-term future of sandy coasts (Cooper et al., 2020; Vousdoukas et al., 2020), there is a growing body of evidence suggesting that they are vulnerable environments and under increasing pressure from rising seas (Vitousek et al., 2017; Almar et al., 2021b), changing wave climates (Allan and Komar, 2001; Reguero et al., 2019; Erikson et al., 2022) and shifts in land use practices (e.g., deforestation, wildfire, increased engineering of coastal/fluvial environments [Syvitski et al., 2022; Warrick et al., 2023]). Such processes that affect coastal morphodynamics add significantly to the uncertainties of future coastal hazards such as flooding and erosion (Barnard et al., 2019).

Over the past few decades, monitoring coastal change has mostly relied on expensive, labor-intensive methods such as aerial, LiDAR or *in situ* measurements, resulting in data sets with limited spatiotemporal resolution. Consequently, only a few studies have been able to consistently monitor coastal morphological changes on regional scales using these methods (Vitousek et al., 2022). The difficulty of collecting coastal morphologic data sets with high spatiotemporal resolution over large scales hinders not only the understanding of coastal change but also the validation of coastal flooding and erosion models.

Publicly available multispectral satellite imagery (MSI) has recently enabled large-scale studies of coastal change (e.g., Luijendijk et al., 2018). Landsat (5, 7 and 8, respectively, launched in 1984, 1999 and 2013) and Sentinel-2 (launched in 2015) missions provide a wealth of open-source images covering the majority of the world's surface (Turner et al., 2021) and are made available through cloud-computing platforms (Gorelick et al., 2017). So far, several studies have adopted MSI for coastal monitoring purposes, focusing on the remote-sensing methodologies (Bishop-Taylor et al., 2019; Vos et al., 2019; Castelle et al., 2021) and coastal change at various scales ranging from global (Luijendijk et al., 2018; Mentaschi et al., 2018) to regional (Vos et al., 2023a) to local (Castelle et al., 2021; Taveneau et al., 2021; Vitousek et al., 2023). Machine learning-based methods have been increasingly applied in satellite-derived shoreline (SDS) extraction algorithms as well (McAllister et al., 2022). The applications of remote sensing for monitoring coastal change have also been facilitated by the development of open-source toolkits, such as CASSIE (Almeida et al., 2021), CoastSat (Vos et al., 2019) and SHOREX (Sánchez-García et al., 2020). These toolkits allow for the automatized extraction of the instantaneous waterline (hereafter referred to as satellite-derived waterline [SDW]), usually in the form of a time series of cross-shore position along user-defined transects perpendicular to the coastline.

The study of coastal change using earth-observing satellites is still an emerging field, and despite the breakthroughs enabled by the availability of satellite imagery and the development of remote-sensing and machine-learning methods, the research community is still facing challenges regarding the extraction of accurate shoreline features (Vos et al., 2023b). First, despite the subpixel methods used in studies (e.g., Bishop-Taylor et al., 2019; Vos et al., 2019) to acquire shoreline position data at a resolution finer than the pixel size, shoreline extraction from satellite imagery still relies on optical imagery with medium pixel resolution (e.g., 10–15 m pan-sharpened pixels), limiting the accuracy to approximately 5–10 m for shoreline products extracted from Landsat and Sentinel missions (Vos et al., 2019; Sánchez-García et al., 2020; Vos et al., 2023b). However, recent research by Doherty et al. (2022) has shown that the accuracy of shoreline products can reach <5 m

using images from PlanetScope missions captured with 3 m resolution imagery. Additionally, rigorous shoreline determination from satellite MSI (hereafter referred to as SDS) typically requires additional data in the form of hindcasted (or observed) waves and water levels. These data sets are crucial to correct the initially extracted SDW data for hydrodynamic processes affecting the instantaneous waterline observed in satellite imagery. A standardized (e.g., based on mean sea level [MSL] or mean high water [MHW]) shoreline measurement reference is paramount for an objective assessment of shoreline variability and for comparison with *in situ* data, which emphasizes the derivation of the best possible SDS data sets from SDW data sets. Water-level corrections are particularly important for beaches subject to high-energy wave climates and/or large tidal ranges (Castelle et al., 2021), though these corrections are less important for low-energy, microtidal coastal environments where the waterline position (SDW) roughly coincides with the MSL/MHW shoreline position (SDS). Processes that drive nearshore water-level fluctuations (e.g., tide and wave runup) add uncertainties in the visual identification of waterline features (Moore et al., 2006), which are typically based on extracting the instantaneous waterline for each image. Lastly, the heterogeneous nature of coasts worldwide complicates the large-scale uses of SDS methods. In 2018, Luijendijk et al. (2018) have reported the first global-scale coastal change study relying on the SDS. Despite its novelty and success, the study also had some limitations. For example, the misidentification of some rocky and fully armored coasts as sandy and the use of composite (time-averaged) images to circumvent uncertainties in instantaneous waterline position and cloud cover (Vitousek et al., 2023) highlight the challenges of applying a single methodology to a wide variety of coastal settings. Similarly, the adoption of multiple indicators for the shoreline, detailed in Boak and Turner (2005), leads to subjectivity in the shoreline detection analysis, particularly when comparing two shoreline change data sets developed using different shoreline proxies/definitions (e.g., MSL vs. MHW contour-based shorelines).

In this study, we seek to address two of the points raised above, that is, (1) the correction of SDW data sets and (2) their applicability for large-scale coastal monitoring. Using the open-source Python (Van Rossum and Drake, 2009) toolkit CoastSat (Vos et al., 2019), we extracted SDW time series along the dissipative beaches of the Columbia River Littoral Cell (CRLC) in the northwestern United States. Beaches along the CRLC are mildly sloped (Ruggiero et al., 2005) and have a large and often complex intertidal foreshore region. Therefore, the waterline identified in satellite imagery is significantly influenced by synoptic variations in water level due to tide and wave runup (Ruggiero et al., 2003). Moreover, the waterline is often hard to distinguish along the CRLC because of the wet sand and the persistence of a thin layer of water on the mildly sloping foreshore topography. Following the methodology of Castelle et al. (2021), we investigated the contribution of wave runup and its components, in addition to tide levels, when applying a series of water-level corrections over the SDW time series. We used all available images from Landsat 5, 7 and 8 missions with 50% or less cloud coverage without any manual image selection or removal protocol. While we did not perform manual quality control on satellite imagery, we found that applying an automated outlier correction to the data set, based on excluding data points greater than a particular factor of median absolute deviations (MADs), was necessary to limit errors when conducting automated shoreline extraction over large scales. The resulting SDS data from this methodology is a dense time series of cross-shore shoreline positions (equivalent to the MHW contour) along the CRLC during

1984–2021 with a spatial (i.e., alongshore) resolution of 50 m and an approximately monthly temporal resolution.

The remainder of this article is organized in the following order: Section "Study area" introduces the CRLC and its characteristics for coastal change monitoring. Section "Methods" describes the methodology to extract the SDW time series and mitigate the errors/biases due to tides, wave runup and outliers. Section "Results" investigates the performance of the waterline position extraction/correction procedure and explores 37 years of coastal change along the CRLC through the lens of high-resolution, satellite-derived MHW shoreline change data. Section "Discussion" discusses the implications of our findings in terms of the capabilities of satellite-based methods for regional-interdecadal coastal change monitoring. Lastly, conclusions are drawn in section "Conclusions."

## Study area

The CRLC is a 165-km-long littoral cell on the US West Coast, extending from Point Grenville, Washington, at its northernmost limit, to Tillamook Head, Oregon, at its southernmost limit. As depicted in Figure 1, the CRLC consists of four mostly sandy subcells (from north to south): North Beach, Grayland Plains, Long Beach and Clatsop Plains. The CRLC surrounds the mouth of the Columbia River, which is responsible for the largest water discharge volume on the US West Coast (Benke and Cushing, 2005; Naik and Jay, 2011). This river system experienced more than a 70% decrease in sand transport since the late nineteenth century (Naik and Jay, 2011) due to its extensive management and regularization

(Gelfenbaum et al., 2001). The littoral cell is characterized by a high-energy wave climate with deep-water significant wave heights and periods having annual averages exceeding 2 m and 10 s, respectively (see Supplementary Figure S1), and a mesotidal range of 2–4 m (Ruggiero et al., 2005). The littoral cell is significantly impacted by El Niño events, the warm phase of El Niño/Southern Oscillation (ENSO), which is a complex climate pattern characterized by the periodic warming of sea surface temperatures in the central and eastern equatorial Pacific Ocean. The most recent strong El Niño events took place in 1982–1983, 1997–1998 and 2015–2016. Under strong El Niño conditions, sea levels and wave activity are significantly impacted, especially during winter, with some storm events leading to extreme wave activity (Allan and Komar, 2002; Barnard et al., 2015, 2017). For example, in February 1998, average winter wave heights were 2 m larger than the typical seasonal conditions along the CRLC (Ruggiero et al., 2005). This increase in wave heights introduces more variations in nearshore water levels via wave setup, exposing coasts to erosion of beaches and dunes (Barnard et al., 2017).

Beach topography surveys have been conducted approximately every 3 months since the summer of 1997, and nearshore bathymetry surveys have been conducted once a year since 1999 along the CRLC (Ruggiero et al., 2005). Northwestern US coasts, including the CRLC, have also been monitored via LiDAR surveys carried out in 1997, 1998, 2002, 2009 and 2016 and through aerial photographs prior to that (Kaminsky et al., 2010; Ruggiero et al., 2013; Mull and Ruggiero, 2014). These studies reveal that beaches along the CRLC are mostly prograding, meaning that the shoreline tends to accrete seaward and, therefore, the beach width often increases

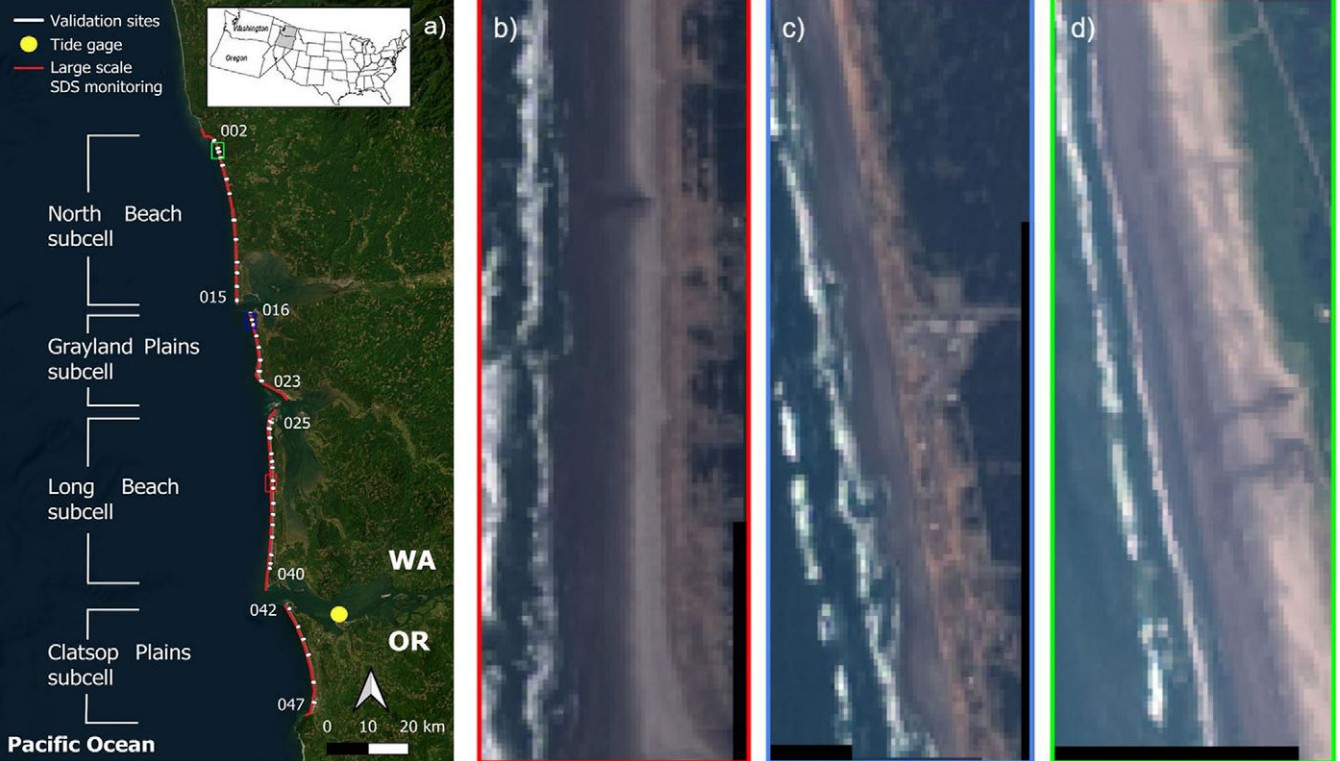

**Figure 1.** The Columbia River Littoral Cell (CRLC). (a) Map of the CRLC showing validation sites where beach profile surveys (white dots) have been conducted (Ruggiero et al., 2005). The labels of some of the transects at the edge of each subcell are displayed. The red line in this panel shows the extent of the areas where satellite-derived coastal change monitoring is conducted for the 1984–2021 period using the presented SDS method. The colored squares show sections of the (b) Long Beach (red), (c) Grayland Plains (blue) and (d) North Beach (green) subcells.

(Ruggiero et al., 2016). According to Ruggiero et al. (2013), beaches along the CRLC are experiencing an average shoreline change rate of +4.2, +1.7, +4.7 and +1.9 m/yr for North Beach, Grayland Plains, Long Beach and Clatsop Plains subcells, respectively, where positive values indicate accretion.

Beaches along the CRLC are relatively flat, where mean beach slopes (represented by tan $\beta$) vary in the range of 0.01–0.05 (Ruggiero et al., 2005). These beaches are mostly covered with sand with a mean grain size of approximately 0.2 mm on the intertidal zone. A small number of beaches in the region have relict coarse sand deposits; for instance, the sand on the mid-beach of the northern Grayland Plains subcell has a diameter that varies between 0.6 and 0.7 mm (Kaminsky et al., 2010). Because of the gentle slopes, high wave energy, and large tidal range, the sand gets wet and darkens quite far landward from the mean waterline. The abrupt transition in the sand color associated with the wet and dry portions of the beach (see, e.g., Figure 1b–d) often makes the dry/wet interface optically very similar to the instantaneous waterline. Note that the flatness of the beaches significantly amplifies this effect, where small variations in water level induce large horizontal variations in the position of the waterline, leaving large portions of the beach wet for hours.

## Methods

### Shoreline position extraction and correction

In this study, the Python toolkit CoastSat (Vos et al., 2019) was used to extract the time series of SDW positions from publicly available satellite imagery products along the CRLC during 1984–2021. Coast-Sat downloads and processes satellite images from Landsat 5, 7 and 8 missions available through Google Earth Engine (Gorelick et al., 2017). All images with <50% cloud coverage are included in the analysis, and no manual image selection/removal was performed on the resulting image collection. Between 1999 and 2019, on average across all transects, 918 images are taken by Landsat 5, 7 and 8 missions, with a cloud coverage <50% (i.e., cloud coverage threshold is set to 50%), meaning a mean rate of 45 images per year (~1 image per week; see Supplementary Figure S2 for the temporal evolution of image availability per each Landsat mission). These images have a pixel size resolution of 30 m. Among these images, the average cloud coverage rate was 19%. The waterline is detected from each image using a pixel-indexing method based on the Modified Normalized Difference Water Index (MNDWI; Xu, 2007), defined as

$$\text{MNDWI} = \frac{G - \text{SWIR}}{G + \text{SWIR}}, \qquad (1)$$

where $G$ and SWIR are the intensity of the green and short-wave infrared bands, respectively. A multilayer perceptron pixel classification method categorizes each pixel into four classes of "sand," "water," "white-water" and "other land features" (Civco, 1993), based on pixel intensity of red, green, blue, near infrared, short-wave infrared bands and their spatial variances, and trained over a set of Australian beaches. For each image, an MNDWI threshold is computed using Otsu's threshold method (Otsu, 1979) that optimally splits the "sand" and "water" portions of the MNDWI histogram, and, finally, the extracted waterline is refined using a subpixel contouring method (Cipolletti et al., 2012). See Supplementary Figures S3 (good cases) and S4 (bad cases) for three-panel figures showing examples of the RGB, pixel-classified and MNDWI-classified images.

Similar to other proxy-based shoreline position estimates (e.g., aerial photograph-derived shorelines; Moore et al., 2006), SDW data are subject to biases and uncertainties due to the presence of tides, waves, atmosphere-induced water-level variations and lighting conditions. The biases/uncertainties associated with shoreline positions are generally low for steep, micro-tidal, low wave-energy beaches. On the other hand, for flat, meso- to macro-tidal, high wave-energy beaches, as is the case along the CRLC, vertical variations in water level can reach several meters and lead to tens of meters of variations in shoreline position. Therefore, in order to obtain an objective comparison of SDS position with *in situ* shoreline observations, typically measured as a water level-invariant elevation contour (or datum-based shoreline, which here is taken to be equivalent to the MHW elevation), it is often necessary to correct the initially extracted SDW for the introduced biases/uncertainties associated with the water level in order to obtain accurate SDS data. These corrections should ideally seek to correct for all hydrodynamic contributions to the waterline time series and retain all of the morphologic contributions to the changing beach location.

In this study, tide and wave runup corrections, based on the 2% exceedance level of wave runup, $R_{2\%}$, have been applied to the SDW time series. Additionally, an outlier correction was used to address anomalies in the time series due to algorithmic misidentification of the waterline. The contribution of these individual corrections is investigated in section "SDS performance." The corrected cross-shore shoreline position (SDS), $X_c$, at time $t$ is given by

$$X_c(t) = X_r(t) + \Delta X_\text{tide}(t) + \Delta X_\text{wave}(t) + \xi(t), \qquad (2)$$

where $X_r$ is the raw cross-shore position of the waterline (SDW), $\Delta X_\text{tide}$ is the tide correction, $\Delta X_\text{wave}$ is the wave (runup) correction, and $\xi$ is an outlier correction term.

### Tide correction and beach slope estimation

Following the approach developed by Vos et al. (2019), SDW data are virtually projected from the instantaneous tide-level elevation to a static, reference elevation $z_\text{ref}$ by considering the beach as a linear, inclined plane that intersects the water surface at an angle noted $\beta$. In this study, all elevations are reported relative to the North American Vertical Datum of 1988 (NAVD88). The reference elevation we use is $z_\text{ref} = 2.1$ m, which roughly corresponds to the MHW elevation along the coasts of Oregon and Washington (Ruggiero et al., 2013). The projection, or vertical correction, of the waterline along this inclined plane to the MHW results in a horizontal tide correction of the shoreline position ($\Delta X_\text{tide}$) defined as

$$\Delta X_\text{tide}(t) = \frac{\eta(t) - z_\text{ref}}{\tan(\beta)}, \qquad (3)$$

where $\eta$ is the tide level (in meters, NAVD88) and tan $\beta$ is the foreshore beach slope. Here, the instantaneous tide levels, with a 1-h temporal resolution, are obtained from the National Oceanic and Atmospheric Administration (NOAA) tide gauge station in Astoria, Oregon (ID 9439040; NOAA, 2022) (see Supplementary Figure S1). CoastSat toolkit contains a package that estimates the foreshore beach slope, tan $\beta$, based on the time series of waterline position (Vos et al., 2020). Foreshore beach slopes are critical to correcting the SDW data for the influence of tides and waves (see equations (3) and (4)). To ensure that the slopes produced via CoastSat correspond best to the *in situ* slopes, we validated and calibrated CoastSat slopes (hereafter denoted by "calibrated slopes") such that they best match with the *in situ*-derived slopes (the validation results are presented in Supplementary Figure S5).

## Wave runup correction

Instantaneous nearshore water levels (and thus waterline positions) are influenced by wave runup, which is the sum of two components: wave setup, the persistent superelevation of nearshore water levels in the presence of breaking waves, and swash, the oscillation of waves washing up and down a beach, which itself has two components of incident band and infragravity band wave swash. Wave runup directly affects the position of the waterline, inducing a cross-shore displacement of the waterline landward (Ruggiero et al., 2003). From a remote sensing point of view, wave runup can also indirectly affect waterline extraction by wetting the sand; for example, a big swash event prior to image collection can wet the beach over tens of meters and thus make it harder to objectively identify the waterline position afterward.

SDW extracted using CoastSat over the CRLC are generally located between the wet/dry sand interfaces (i.e., edges delineated by the last tide and the maximum wave runup levels) and the instantaneous waterline. Therefore, we investigate the contribution of extreme wave runup maxima, $R_{2\%}$, to mitigate this bias in the estimation of the shoreline positions. The wave-induced shoreline cross-shore position bias is defined as

$$\Delta X_{\text{wave}}(t) = \frac{R_{2\%}(t)}{\tan(\beta)}, \tag{4}$$

where $R_{2\%}$ is the 2% exceedance runup (Moore et al., 2006; Senechal et al., 2011), which has been parameterized by Stockdon et al. (2006) as

$$R_{2\%}(t) = 1.1\left(0.35\tan(\beta)\sqrt{H(t)L(t)} \right. \tag{5}$$

$$\left. + \frac{\sqrt{H(t)L(t)(0.563\tan^2(\beta)+0.004)}}{2}\right),$$

where $H$ is the wave height, $L$ the wavelength and $\beta$ the beach slope angle (in radians). This relationship for runup accounts for elevation variations due to wave setup, $R_{\text{setup}}$, incident swash, $R_{\text{incswash}}$, and infragravity swash, $R_{\text{igswash}}$. These three components, which have been used separately at some point in this study, reshape the estimation of $R_{2\%}$- as follows:

$$R_{2\%}(t) = 1.1\left(R_{\text{setup}}(t) + \frac{\sqrt{R_{\text{incswash}}(t)^2 + R_{\text{igswash}}(t)^2}}{2}\right). \tag{6}$$

The wave data set utilized in our analysis is from the CAWCR/CSIRO wave hindcast product (Durrant et al., 2019), which provides wave height, period and direction in deep water offshore of the Columbia River at a 1-h temporal resolution (see Supplementary Figure S1). This wave hindcast data set has been validated against altimeters and buoy observations, and the validation metrics show satisfactory agreement between the hindcast and the altimetry/buoy data (Durrant et al., 2014; Smith et al., 2021).

## Outlier correction

SDW data typically contain outliers stemming from various sources (e.g., isolated clouds/fog, geo-referencing, "sand"-"water" segmentation errors, wave effects, etc.). Unmasked clouds appear to be the primary source of misdetection of the waterline, resulting in large, easily identifiable outliers. Other misdetections appear during some low tides, leading to the detection of a continuous waterline near the wet/dry sand interface and also isolated and irregular waterline contours that may intermittently appear along the wet beach. Also, the CoastSat pixel classification sometimes fails to identify wet sand as "sand," resulting in a waterline detected between the instantaneous waterline and the wet/dry sand interface (images illustrating these cases are shown in Supplementary Figure S4). Because the water-level corrections for tide and wave runup can be relatively large compared to the morphological changes of the coast, the outlier correction, used here, is applied after the wave runup and tide corrections in order to avoid the miscorrection of "well-extracted" waterline positions (SDWs).

The outlier correction is conducted on the time series extracted at each transect using the Python package *hampel* (Pedrido, 2021) and a function of the same name. As input, this function takes a window size $S$, a threshold factor $n$ and a time series $X(t)$ and corrects the values identified as outliers based on how much they deviate from a particular multiplication of the median absolute deviation (MAD) of the neighboring data calculated as $\mu$ = median $(X_i-\text{median}(X_S))$. The value $X_i$ is identified as an outlier if its deviation from the S-sized rolling median exceeds $kn$ times the MAD, with a scaling factor conventionally used as $k = 1.4826$ for a normally distributed set of values (Rousseeuw and Croux, 1991). Values identified as outliers are imputed with the rolling median value.

The sensitivity of SDW data to the value of inputs $n$ and $S$ has been investigated and is shown in Supplementary Table S1. The best *RMSE* and $R^2$ scores are found for $n = 1$ and $S = 15$. For the rest of the study, outlier correction refers to the corrections made using the function *hampel* with $n = 1$ and $S = 15$ as input parameters. Supplementary Figure S6 shows examples of waterline position time series before and after outlier correction. All of the above-mentioned corrections applied to the SDW time series are summarized in Table 1.

## Validation of the satellite-derived data

Quality assessment of the resulting satellite-derived data has been made possible based on the comparison with (1) beach elevation profile surveys conducted quarterly for 42 sites (an example is shown in Supplementary Figure S7) along the CRLC between winter 1999 and fall 2018 (Ruggiero et al., 2005) and (2) the 1980s–2002 (1967–2002 for the Clatsop Plains subcell) shoreline change rate product from Ruggiero et al. (2013), which are end-point rates calculated from shorelines (corrected using approaches similar to those adopted in this study) extracted from aerial photographs taken in the late 1960s (Oregon) and late 1980s (Washington) and a LiDAR survey carried out in 2002. The former is used to examine the accuracy of the extracted SDS time series and investigate the influence of the water-level (hydrodynamic) correction procedure on the accuracy of the SDS data, while the latter is used to assess the accuracy of the SDS change trends (rates) extracted at a 50-m resolution all along the CRLC during 1984–2002.

During low tides, beach elevation profiles (with vertical accuracy <10 cm) are measured by walking from the dunes to the sea along each predefined transect while carrying a GPS receiver and antenna mounted to a backpack following the method of Ruggiero et al. (2005). For each profile, the MHW position is extracted from the elevation profile at 2.1 m elevation (NAVD88). Beach slopes can also be extracted from profile data, as we calculated the slopes between MSL and MHW elevations (i.e., around 1.1 and 2.1 m NAVD88, respectively) as the ground truth beach slopes.

**Table 1.** Correction procedure applied to the SDW time series

| Correction label | Correction performed | Terms remained to be corrected |
|---|---|---|
| 1 | Raw data | $\Delta X_{tide} = 0$, $\Delta X_{wave} = 0$, $\xi = 0$ |
| 2 | Outlier removed | $\Delta X_{tide} = 0$, $\Delta X_{wave} = 0$ |
| 3 | Tide corrected from 2 | $\Delta X_{wave} = 0$ |
| 4 | Wave setup corrected from 3 | $R_{incswash} = 0$, $R_{igswash} = 0$ |
| 5 | Incident swash corrected from 4 | $R_{igswash} = 0$ |
| 6 | Infragravity swash corrected from 5 | ALL CORRECTIONS APPLIED |

*Note:* The labels used for each correction are the same used in Figure 2. Note that for corrections 2 through 6, outlier correction is applied at the end, that is, after hydrodynamic corrections.

### Large-scale shoreline position extraction along the CRLC

The process of extracting shoreline positions from satellite imagery is automated within an area of interest (AOI) containing all of the transects on which shoreline positions are sought. AOIs are subdivided into rectangular regions of interest (ROIs) of approximately 25 square km each with an overlap of 400–500 m between the ROIs to ensure that transects on the extremities are entirely contained in at least one ROI. Each transect within each ROI is then processed with the same method as explained above.

In our analysis, transects serve as the alongshore spatial discretization, or the model "grid," spaced 50 m apart from each other sequentially. They extend from an onshore point (i.e., where the cross-shore position is set to zero), which specifies the onshore boundary where the sandy beaches meet vegetation, dunes, bluffs, cliffs or development, toward an offshore point while perpendicular to the shoreline. This shoreline is identified visually via the latest available satellite imagery on Google Earth Pro (Google, 2022) and acts as a "reference" shoreline in defining the transects. The length that the transects are extended beyond the reference shoreline is user-defined and, in our case, is 300 m. This process generates 2,852 transects along the CRLC. Note that the transects used for the large-scale SDS extraction via CoastSat are different than the 42 transects (with much larger inter-transect spacing) where the quarterly beach elevation profile surveys have been conducted since 1997 (Ruggiero et al., 2005).

### Results

#### SDS performance

To demonstrate an example of the evolution of SDS accuracy, we quantified validation metrics at a single profile (transect) along the Grayland Plains using $R^2$, $RMSE$, $\sigma$ and *bias* scores, displayed in Figure 2, where the different corrections described in Table 1 are applied. Figure 2 demonstrates distinct improvement of accuracy scores where a significant decrease in $RMSE$ and *bias* for the outlier removal, tide, wave setup and wave swash (incident) corrections (correction labels 2 and 5) is observed, while a large increase in $RMSE$ and *bias* for wave swash (infragravity) correction (correction label 6) emerges.

To determine the large-scale accuracy of the SDS time series, the derived SDS data (i.e., the corrected SDW data) are validated against the time series of *in situ*-measured shoreline positions at 42 transects along the CRLC; the exact same transects that were used for the extraction of SDS data over these 42 sites. Figure 3 depicts a map of the locations of the beach elevation profiles color-coded according to their coefficient of determination ($R^2$), root mean square error ($RMSE$), standard deviation ($\sigma$) and *bias*, obtained via comparing SDS data against the *in situ* shoreline position data. The mean scores obtained for these 42 locations are 0.54, 22.37, 19.3 and 3.54 m after applying the tide, wave setup and wave swash (incident) corrections and outlier removal (i.e., correction label 5 in Table 1), which were introduced in section "Methods." Spatial variability in Figure 3 is evident; for instance, $R^2$ scores are relatively higher at the extremities of each subcell (e.g., southern Grayland Plains and northern Long Beach), which is likely caused by the large shoreline change trends at these sites.

Long-term SDS trends, $dX/dt_{(SDS)}$, extracted from the outlier-only-corrected SDS time series (correction label 2 in Table 1), demonstrate a strong correlation with *in situ* shoreline change trends, $dX/dt_{(in\ situ)}$, with a coefficient of determination $R^2 = 0.99$ and a $RMSE$ and *bias* <1 and $-0.23$ m/yr, respectively (see Supplementary Figure S8b). The beach slopes initially derived from CoastSat (shown in Supplementary Figure S5a) also show a satisfactory fit with slopes derived from *in situ* profile data, where the coefficient of determination is $R^2 = 0.77$. However, it appears the slope estimation methods in CoastSat slightly overestimate the beach slopes derived from *in situ* data, and the amount of overestimation is larger for steeper slopes. Using the y-intercept and the gradient of the fitting line, we perform a calibration. After calibration, the calibrated CoastSat-derived slopes and the *in situ* ones are fitted on the 1:1 line (see Supplementary Figure S5b).

Figure 4 depicts the evolution of $R^2$, $RMSE$, $\sigma$ and *bias* scores for each correction label averaged across all 42 validation sites for both raw (uncalibrated) and calibrated CoastSat-derived beach slopes. By comparing the two (i.e., using the calibrated vs. uncalibrated slopes), it is apparent that the calibrated slopes (which are generally smaller compared to the uncalibrated ones) typically drive the SDS data further landward under tide correction (see equation (3)) compared to the ground truth data (i.e., bias<0 m after tide correction), which allows the wave correction terms (i.e., wave setup and incident swash) to successively improve the validation metrics, where bias~0 m after applying wave corrections (excluding infragravity wave swash). Corrections to SDW data using local beach slopes extracted from the beach elevation profiles rather than the calibrated SDS-derived beach slope estimation (described in section "Methods") are also performed, with both a static slope, that is, an average of the beach slopes across all profiles, and a dynamic slope, that is, the beach slope extracted from the measured beach elevation profile at the time of each survey. Corrections using the static beach slopes result in the tide, wave setup and wave swash (incident)-corrected SDW data with almost no biases (bias~0 m) but do not significantly affect the other validation metrics ($R^2$~0.5 and RMSE~20 m; see Supplementary Figure S9). As noted in Castelle et al. (2021), we also find that the use of time-varying (i.e., dynamic) beach slopes rather than a static beach slope does not lead to a notable improvement in the accuracy of corrected SDW data (see Supplementary Figure S10).

In terms of $R^2$, $RMSE$, $\sigma$ and *bias* scores, it appears that the outlier, tide, wave setup and incident wave swash corrections systematically provide an increase in SDS data accuracy, regardless of the beach slopes used during the correction procedure, that is, the beach slopes extracted from SDS time series (calibrated CoastSat slopes) or from the beach elevation profiles (static and dynamic).

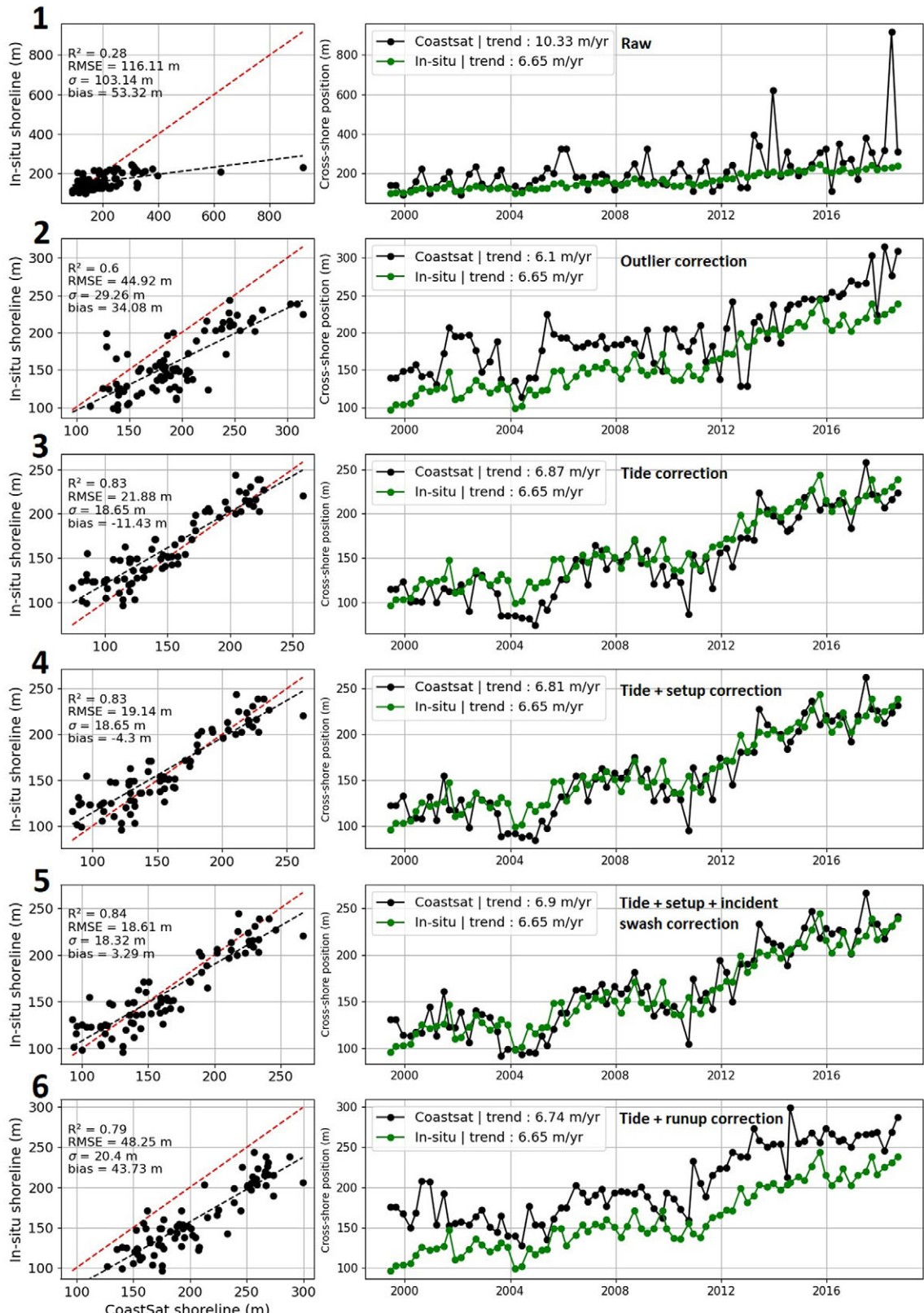

**Figure 2.** Validation plots showing the comparison between satellite-derived and *in situ* shoreline positions (Ruggiero et al., 2005) for a single cross-shore profile (profile 020) at Grayland Plains, Washington. Each row of panels ranging from 1 to 6 refers to its corresponding correction label described in Table 1. On each row, the right and left panels represent the direct comparison of satellite-derived and *in situ* shoreline positions and the shoreline position time series from CoastSat and *in situ* measurements, respectively. $R^2$, *RMSE*, $\sigma$, *bias* and long-term trends are shown for each correction label. Red dashed lines in direct comparison subplots show the 1:1 line, and black dashed lines show the linear regression between the satellite-derived and *in situ* shoreline position data. Note that the higher positive values correspond to the seaward direction.

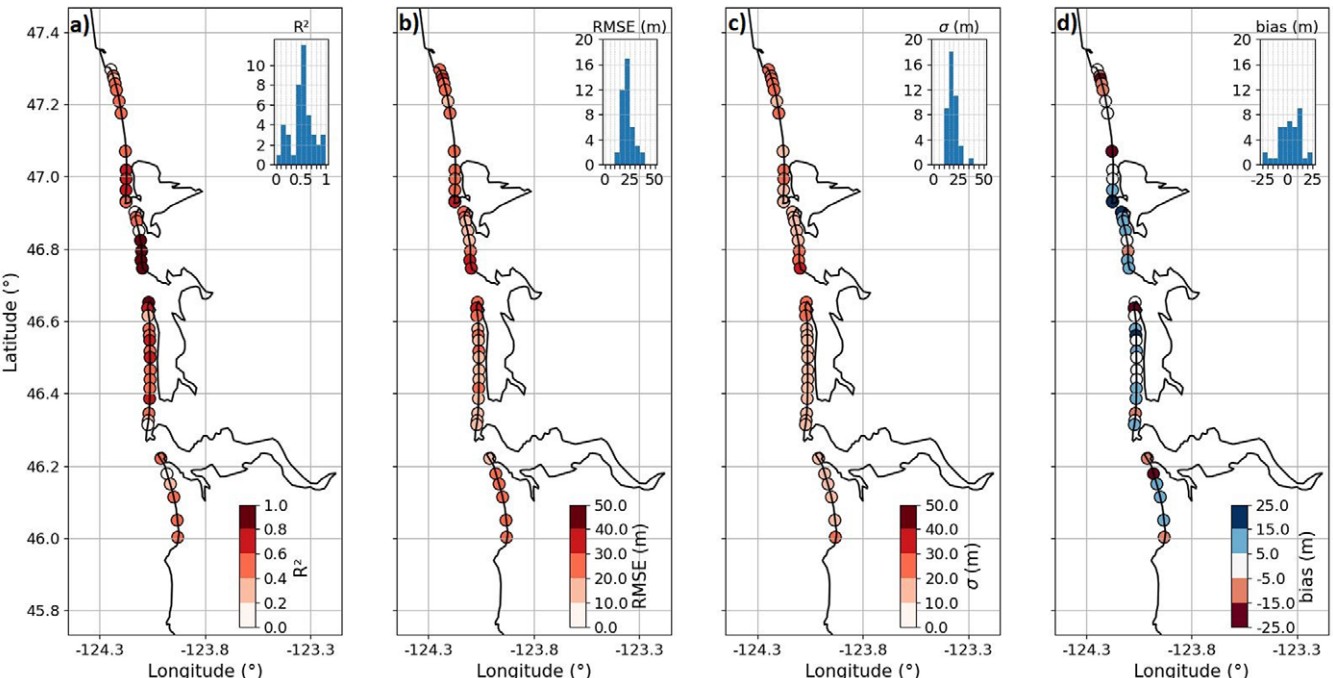

**Figure 3.** Maps of the CRLC showing the 42 validation sites and the validation scores, that is, (a) coefficient of determination ($R^2$), (b) root mean square error (*RMSE*), (c) standard deviation ($\sigma$) and (d) *bias*, between the time series of cross-shore shoreline position extracted via CoastSat (corrected for tide, wave setup, wave swash [incident] and outliers) and the *in situ*-measured beach elevation profiles (Ruggiero et al., 2005). On each subplot, the histogram shows the distribution of validation scores. Relatively accurate estimations of the long-term trends are shown in Supplementary Figure S10b.

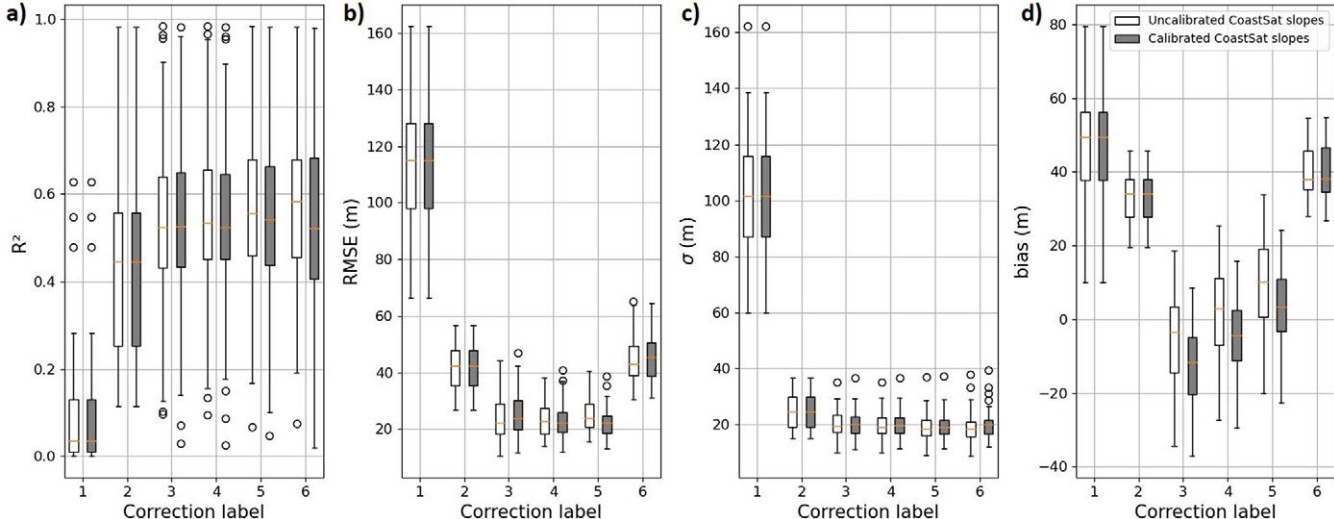

**Figure 4.** Box plots showing (a) $R^2$, (b) *RMSE*, (c) $\sigma$ and (d) *bias* scores after each step of correction for the 42 sites along the CRLC. Corrections steps, labeled with numbers 1 to 6, are described in Table 1. Black circles depict the values out of the range of box plots. White boxes (gray boxes) indicate validation scores for corrections performed using uncalibrated beach slopes (calibrated against *in situ* beach slopes) calculated via the CoastSat toolkit (Vos et al., 2019).

Thus, partial application of wave runup correction by excluding the infragravity wave swash is most advantageous in CRLC (also noted in Cabezas-Rabadán et al., 2020). However, the outlier correction-only provides modest validation scores ($R^2 \sim 0.45$, *RMSE* $\sim 30$–50 m), but panels (a) and (d) in Figure 5 show the shoreline change trends during 1984–2002 calculated via linear regression over only the outlier-corrected SDWs compared to the endpoint shoreline change rate product from Ruggiero et al. (2013) introduced in section "Methods". The fit between the two data sets ($R^2 = 0.74$) shows that the SDS method is capable of extracting high-quality

shoreline change trends over large scales. In addition, the differences between SDS trends and end-point trends may stem from subtle differences in rate calculation methodology, that is, differences between linear regressions and endpoint rates.

### 37 years of morphological change along the CRLC

We examined the temporal evolution of the shoreline positions at all 2,852 transects along the CRLC and also their deviation from the long-term shoreline change trend from 1984 to 2021. Figure 6a shows

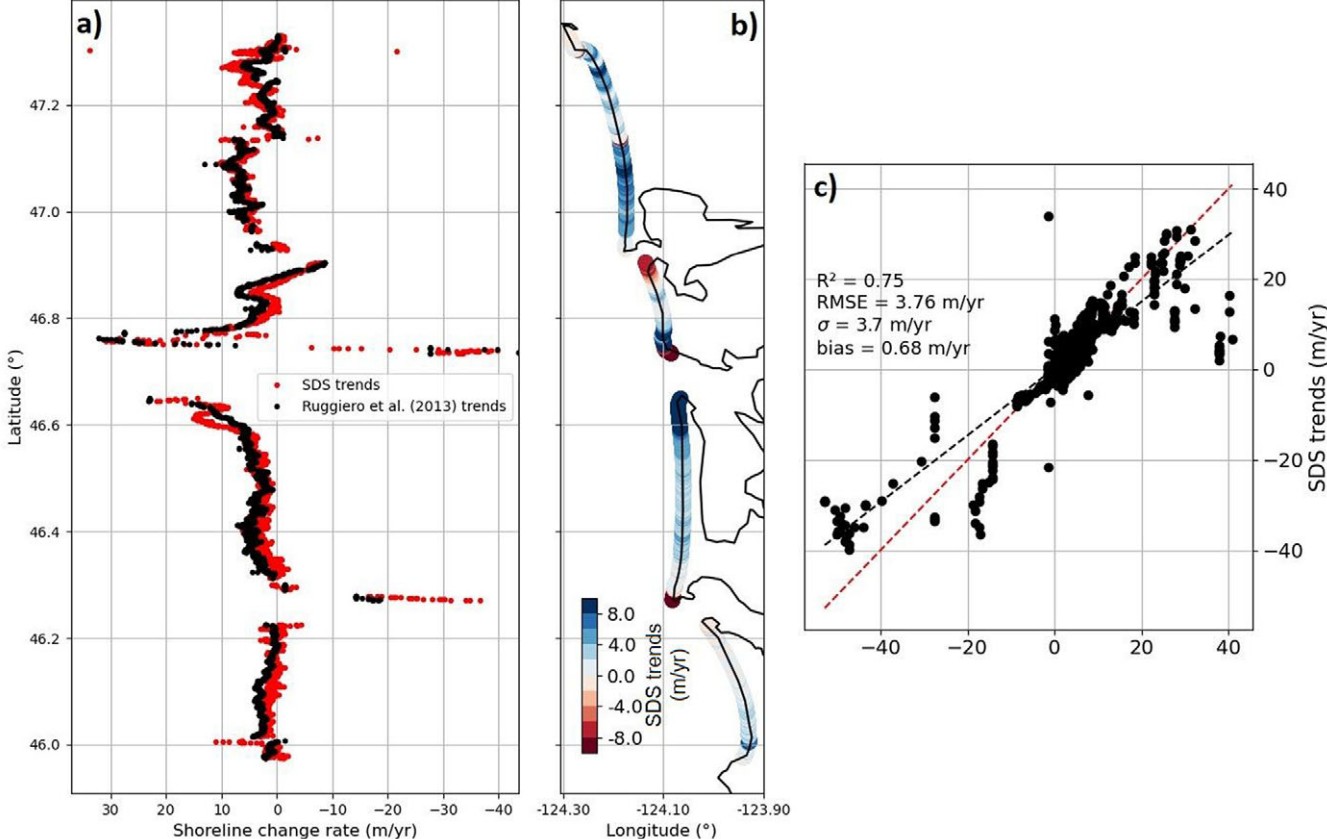

**Figure 5.** Shoreline change trends along the CRLC. (a) Latitudinal (alongshore) variability of 1980s–2002 (1967–2002 for the Clatsop Plains subcell) shoreline change trends from Ruggiero et al. (2013) in black, and 1984–2002 SDS change trends in red, (b) distribution map of SDS change trends along the CRLC and (c) direct comparison between shoreline change trends from Ruggiero et al. (2013) and satellite-derived data.

the spatiotemporal evolution of the shoreline positions along these transects during 1984–2021. It highlights that CRLC beaches have generally experienced accretion over the past four decades. Shoreline change rates are heterogeneously distributed among the subcells; for instance, the Clatsop Plains subcell (south of the Columbia River mouth) experiences lower shoreline change rates than the other three subcells north of the Columbia River mouth, and beaches at the edges of the subcells are generally experiencing erosion. Moreover, Figure 6b depicts the evolution of anomalies (deviations) in shoreline position relative to the local shoreline change trend, calculated at each transect via

$$X_{\text{anom}}(t) = X(t) - X_{\text{trend}}(t), \tag{7}$$

where $X(t)$ is the change in tide-corrected shoreline position time series (in meters) since spring 1984 (time series start from zero), and $X_{\text{trend}}(t)$ is the linear trend line fit to the observed shoreline position time series, as shown in Figure 6.

This spatiotemporal visualization of shoreline position anomalies reveals seasonal-to-interdecadal shoreline change patterns; seasonal cycles are revealed with "high-frequency" vertical stripes, while a longer-term, "low-frequency" pattern tends to indicate that the CRLC roughly experienced an increase in accretional trends relative to the long-term linear rate during 1984–1996, a decrease during 1997–2014 and an increase again since. It is worthwhile to point out the similarity in the temporal evolution of shoreline position anomalies between distant locations, such as the North Beach subcell and the Long Beach subcell, which are dozens of kilometers apart.

To assess the potential link between the "low-frequency" pattern in shoreline positions shown in Figure 6b and the modes of climate variability in the Pacific Ocean, we examined the temporal evolution of the Pacific decadal oscillation (PDO; Mantua et al., 1997) and Niño 3.4 (Rayner et al., 2003) indices. Figure 6c shows the temporal evolution of PDO, which is a long-term climate pattern that involves variations in sea surface temperatures in the North Pacific Ocean. In this figure, the time series of PDO (in dashed gray) and the scaled mean anomaly of cross-shore shoreline positions along the CRLC (in solid black) are displayed during 1984–2021. The figure highlights that these time series seem to roughly evolve jointly, albeit this correspondence is more evident for specific periods, for example, the 1997–2013 relative erosional pattern coincides with the negative PDO phases, and, reversely, 1984–1996 and 2014–2021 relative accretional patterns coincide with the positive PDO phases. We observe that the 2014–2017 evolution of the PDO greatly matches the signal of median shoreline anomaly change. Similar to Figure 6c, the time series of Niño 3.4 index (in dashed gray) and the scaled mean anomaly of cross-shore shoreline positions along the CRLC (in solid black) are shown in Figure 6d. Niño 3.4 is an index used to monitor and quantify the strength of El Niño and La Niña (opposite phase to El Niño) events through measurements of sea surface temperature anomalies in the Pacific Ocean. Large positive values of Niño 3.4, which characterize major El Niño events, do not seem to drive drastic erosional events as have been observed in shoreline change studies in California (e.g., Barnard et al., 2017). The positive peak in 1997–1998 matches

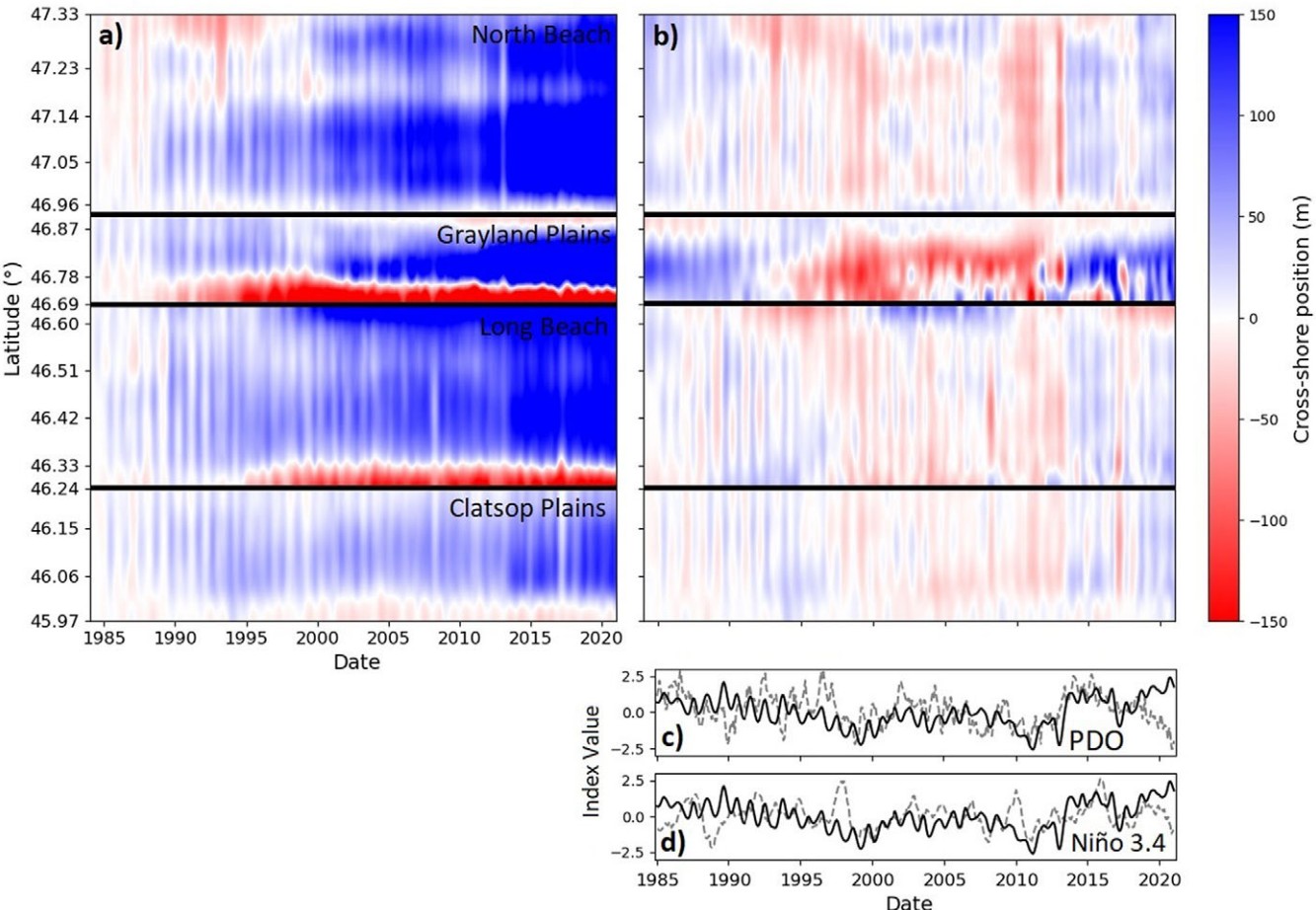

**Figure 6.** Spatiotemporal evolution of (a) the cross-shore shoreline position along the CRLC, initially set at 0 m, and (b) the associated anomalies relative to the local shoreline change trend during the same period, and temporal evolution of (c) PDO and (d) Niño 3.4 indices during 1985–2021 (Mantua et al., 1997; Rayner et al., 2003) in gray dashed lines. The mean anomaly of cross-shore shoreline position at all transects scaled by its standard deviation is also displayed on panels (c) and (d) (black solid lines). Note that the positive (negative) values of PDO and Niño 3.4 metrics correspond to El Niño (La Niña) conditions. Panels (a) and (b) are generated using the MATLAB (2010) interpolation/smoothing function *smoothn* (Garcia, 2023). Each subcell has been processed and smoothed separately with the same parameters and then aggregated together.

the 1997–1998 winter negative peak of the scaled mean anomaly of cross-shore shoreline position along the CRLC, but does not seem to significantly influence the winter erosion pattern. The same pattern can be observed for a smaller positive peak in the Niño 3.4 index in 2009–2010. Similar behavior is observed for the 2015–2016 ENSO event, which corresponds to the largest (positive) Niño 3.4 index value during the study period. Moreover, the strong negative peak of Niño 3.4 in 1988 did not result in any significant change in the anomalies of the cross-shore shoreline positions along the CRLC.

## Discussion

### Applications of spatiotemporally high-resolution SDS data for coastal management

This study demonstrates the applicability of SDS methods for local-to-regional and monthly-to-interdecadal scale coastal change analysis over the high wave-energy, dissipative sandy beaches of the CRLC. SDS methods can potentially be applied to a broad range of coastal environments, particularly those experiencing high shoreline change trends. This study also demonstrates that satellite monitoring can detect spatial and temporal shifts in erosion/

accretion hot spots and can greatly increase the scale of observations, while *in situ* field measurements are restricted by their local scale and temporal infrequency. The use of SDS methods can thus greatly reduce the costs associated with coastal monitoring operations over large spatiotemporal scales since only a few field measurements are needed to validate SDS data once it is established that the SDS method used is appropriate for the application site/region.

Current SDS toolkits, such as CoastSat, support the quick generation of high-resolution shoreline position data sets over large spatiotemporal scales. Over large scales, the signal-to-noise ratio for dissipative beaches, that is, the ratio of low-frequency shoreline change and high-frequency biases due to tides and wave runup, allows us to conduct reliable estimates of erosion/accretion trends, even without water-level corrections. However, obtaining accurate SDS time series generally requires corrections for tides and outliers, at the very minimum.

Strong variability of the shoreline, as experienced by the Grayland Plains subcell, can expose local coastal areas to a high level of vulnerability. Along the CRLC, the PDO signal, which was on average positive during the 1984–1997 and 2014–2021 periods and negative during 1997–2014, seems to lead to a "low-frequency" signal of the anomalies of the shoreline change trends with an amplitude >10 m, and locally exceeding 50 m along the Grayland

Plains subcell. Some very recent studies have investigated the link between coastal morphologic change and climate modes, such as the occurrence of major El Niño events (Barnard et al., 2017; Anderson et al., 2018; Almar et al., 2023; Vos et al., 2023a), quantified through metrics such as the Multivariate ENSO Index (MEI) (Wolter and Timlin, 1993) and Niño 3.4 (Rayner et al., 2003). Vos et al. (2023a) have found that beaches along the US West Coast (i.e., California beaches) generally experienced erosion (accretion) during the boreal winter El Niño (La Niña) phases. Our preliminary results suggest that the CRLC does not respond as strongly as California does to boreal winter El Niño phases (see Figure 6d). However, establishing a clear link between the inter-annual shoreline position anomalies along the CRLC and ENSO cycles is beyond the main scope of this study and could benefit from further research, not merely along the CRLC but the entire Pacific Northwest (PNW).

Historical SDS positions have emerged as revolutionary in the modeling of future shoreline positions by driving coastal science toward a "data-rich" field (Vitousek et al., 2023). In addition to supporting large-scale trend analyses, SDS data sets can greatly benefit dynamic shoreline modeling efforts. Various coastal geomorphology evolution models (e.g., Vitousek et al., 2017; Ibaceta et al., 2020; Taherkhani et al., 2023), particularly those applying data assimilation techniques, rely on historical shoreline data to calibrate their free parameters, which can be highly location/time-dependent. While field- and airborne-derived shoreline positions come with very high precision ($RMSE < 1$ m), shoreline positions obtained via these methods are typically very sparse throughout time (e.g., often annual at best and often with resampling > 5 years), mainly due to being expensive, rendering the calibration/training of these models insufficient. Nevertheless, incorporating SDS data sets in coastal evolution models has significantly enhanced the efficiency of the calibration process and reliability of the projected short- and long-term future shoreline positions (e.g., CoSMoS-COAST; Vitousek et al., 2023). Ultimately, future advances toward more accurate detection of SDS data sets (via better correction and manipulation of SDW data sets) can lead to enhanced model calibration and, thus, more robust future shoreline projections.

### Accuracy, reliability and limitations of SDS data

The cross-shore shoreline position time series extracted along the dissipative beaches of the CRLC (using the procedures described in section "Methods") show medium accuracy when validated against the ground truth data (on average: $R^2 \sim 0.55$, $RMSE \sim 20$ m), relative to the best results that SDS methods can achieve in other idealized settings (Bishop-Taylor et al., 2019; Doherty et al., 2022), including using Landsat/Sentinel images only (Vos et al., 2019). Castelle et al. (2021) have shown that high-quality shoreline data ($R^2 > 0.8$, $RMSE \sim 10$ m) could be extracted along dissipative beaches by applying both tide and wave runup corrections and also by manually removing flawed waterline detection (Castelle et al., 2022). Although their method is efficient for the generation of reliable shoreline data, the manual selection of images is onerous for large-scale shoreline position extractions. Moreover, the wave runup correction they used (Senechal et al., 2011; Castelle et al., 2021) has been specifically calibrated for their study site. These conditions for accurate shoreline extraction along high-energy, meso/macro-tidal beaches are difficult to meet in regional-to-continental coastal change studies, as the one conducted by Vos et al. (2023a) over the Pacific Basin, where the PNW region has been left out of the analysis due the challenges associated with processing satellite-derived images of this coastal region. Therefore, future work could enhance water-level corrections, develop criteria and methods for the automatic removal of flawed images and explore outlier correction methods that mitigate biases while preserving the integrity of the shoreline position time series.

Additionally, it seems that CoastSat sometimes captures virtual waterline features between the wet/dry sand interface and the actual instantaneous waterline along the CRLC. The resulting noise is likely to perturb the results of frequency-domain mode analysis, consequently affecting estimations of beach slope via CoastSat. It must be noted that an overestimation of the beach slope, as observed along the CRLC, leads to an underestimation of water-level corrections (especially tide correction, as shown in Figure 4 when using the uncalibrated CoastSat-derived slopes). This tendency of confusion in the detection of the waterline, which has also been observed along southwestern French beaches (Castelle et al., 2021), might be due to the fact that wet sand turns dark and contrasts greatly with dry sand, leading to the formation of an easily observable interface between the wet and dry sand, which is optically very similar to a waterline interface. This highlights the remaining limitations of SDS methods and, thus, the need to validate and jointly consider SDS data with alternative data sources from complementary methods, such as other remote-sensing methods, field surveys and numerical modeling, to ensure the higher accuracy of extracted SDS data.

It is also important to note that the use of shoreline position as a proxy for coastal geomorphology state has its limitations, as it represents only a one-dimensional (1D) land/sea interface and does not capture the complexity of coastal morphodynamics occurring between the lower shoreface and the dunes, highlighting the opportunity for future research to develop methods to quantify and monitor coastal change beyond a 1D shoreline position (e.g., satellite structure-from-motion).

### Toward new ways to monitor coastal change via satellite products

The study presented here fits into a substantial body of research using MSI to monitor coastal change at local and regional scales, with each successive study providing potential improvements for satellite-based coastal change analysis at a given coastal setting. The collaborative advancement of satellite coastal monitoring not only enhances our understanding of the diverse range of beach dynamics worldwide but also incorporates this diversity to progress toward broader-scale studies, for example, robust global-scale analyses of coastal evolution over the past four decades.

Recently, Bergsma et al. (2021) have developed integrated approaches for estimating beach topography and nearshore bathymetry using satellite imagery. Their methods use wave dispersion theory to extract nearshore bathymetry and stereography methods to estimate topography, which involves analyzing stereopairs of high-resolution satellite images to determine beach surface elevations. Nearshore bathymetry is calculated using wave kinematics extracted from satellite imagery and linear, shallow-water wave dispersion theory to derive water depth. Other methods allow for estimating nearshore bathymetry based on light penetration and reflection in the water (Li et al., 2021; Al Najar et al., 2022). These prototypes (e.g., S2Shores [Almar et al., 2021a] and SaTSeaD [Palaseanu-Lovejoy et al., 2023]) have matured over the past few years and now enable the generation of a topography-bathymetry continuum with an error in the vertical position of ~ 1 m, down to ~ 10 cm for ideal cases. However, these methods are still subject to limitations, such as the lack of texture for certain types of beaches,

which may bias the stereography (Taveneau et al., 2021) and the inability to observe waves or extract their characteristics on some images, leading to errors in bathymetry extraction. Despite these challenges, developing these new tools represents a promising path toward using more comprehensive indicators, rather than primitive ones such as shoreline positions, to monitor coastal change.

## Conclusions

In this study, we monitored changes in shoreline positions along the sandy beaches of the CRLC during the past four decades using CoastSat, an open-source SDS extraction toolkit. Based on *in situ* beach profile and ground truth data across the littoral cell, we investigated the contributions of tide and wave runup corrections to reduce errors and biases in the SDS position data. The identification and correction of these errors/biases revealed that tide, wave setup and incident band wave swash corrections and outlier removal systematically improved the accuracy of SDS data at our study site, leading to final outputs with RMSE~20 m using Landsat 5, 7 and 8 imagery (i.e., *RMSE* of less than a pixel size [~30 m]), where infragravity band wave swash corrections significantly decreased the accuracy. From the tide and outlier-corrected SDS time series, we developed a high-resolution (~50 m-monthly) shoreline position data set along the CRLC during 1984–2021 and found that it provides a reliable and coherent picture of both long-term shoreline trends (with $R^2 = 0.99$, *RMSE* < 1 m/yr) and fine-scale shoreline response to seasonal, interannual and decadal variations in the wave and water level climate.

**Open peer review.** To view the open peer review materials for this article, please visit http://doi.org/10.1017/cft.2023.30.

**Supplementary material.** The supplementary material for this article can be found at http://doi.org/10.1017/cft.2023.30.

**Data availability statement.** The satellite-derived MHW shoreline data set generated and analyzed in this study is available in the Zenodo data repository: https://zenodo.org/uploads/10136946. Beach elevation profile data from the *in situ* monitoring program can be found at https://nvs.nanoos.org/BeachMapping. The CoastSat toolkit used in this study to download and process satellite imagery can be found at https://github.com/kvos/CoastSat.

**Acknowledgments.** We thank Mitchell Harley and the two other anonymous reviewers, and the editor Dr. Kristen Splinter, for their recommendations and insights, which helped us improve the manuscript significantly. Any use of trade, firm or product names is for descriptive purposes only and does not imply endorsement by the US government.

**Author contribution.** M.G. and P.R. developed the initial concept for this study. M.G., M.T. and M.L. processed the data. M.G. performed and verified the analysis. M.G. and M.T. wrote the original manuscript. All authors discussed the results and edited the manuscript.

**Financial support.** This work was supported by the Cascadia Coastlines and Peoples Hazards Research Hub, an NSF Coastlines and People Large-Scale Hub (NSF #2103713). Additional support was provided by NOAA via the NOS/NC-COS/CRP Effects of Sea-Level Rise (ESLR) Program (award number NA19NOS4780180).

**Competing interest.** The authors declare no competing interests.

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
