## [Reviewer Report]

This manuscript implemented the CoastSat for dissipative beaches. By validating the SDS data against in-situ beach survey data, this manuscript discussed the impact of different SDS correction methods including outlier correction, tide correction, wave-setup correction and swash correction. The results showed that only the tide and outlier correction are needed to achieve the best SDS accuracy.

In general, the manuscript is well-written with clear structure and description of methodology with high-quality figures. However, the description for the research implication is insufficient and additional analysis is recommended to clarify some uncertainties related to the SDS algorithm. I, therefore, suggest a major revision.

Major comments

1. As mentioned by the author and also pointed by Mao (2021), many SDS algorithms tend to extract the wet-dry sediment line instead of the water line. I agree with the author that this can be the potential cause to the landward offset of shorelines after all corrections applied however, if the SDS is already on the intertidal zone instead of the instantaneous water line, then the landward offset problem is caused by the over correction of tidal impact instead of the wave-setup. I would like to see how the correction results will change if we assume the SDS was inherently follows the MHW line (i.e. keeping the wave-setup and swash corrections but remove or modify tidal correction).

2. I missed a flowchart to demonstrate (and label) the steps. Currently the author only included a table in supplementary, which is not ideal.

3. In Section 2, I suggest adding the information about the quality and quantity of satellite images across the study period in the study site. This should include the number of L5/L7/L8 images and the statistics of cloud cover rate.

4. This research calibrated the shoreline position to the MHW but the author did not mention this until section 3.1.1. I suggest clarifying the definition of shoreline position related to datum in abstract and introduction.

5. The outlier correction requires sensitivity analysis which relies on the survey data. The author should mention it as a limitation and if possible, shows criteria (e.g. relating the parameters to the statistics of time series) of selecting outlier correction parameters so it can be widely applied without field data.

6. When looking at Figure S2, a large portion of data was identified as outliers. Does it suggest the SDS is very unreliable in this study site? Given the bad quality of input time series, the outlier correction method worked as a median kernel which smoothed the time series but is not helpful to retrieve the real instantaneous value. In this case, it works more like Luijendijk et al., 2018 and Hagenaars, 2018 which used a moving window to smooth image and retrieve the shoreline. I suggest the author comparing their methods and results with Luijendijk et al., 2018 to demonstrate the advantage of the proposed workflow.

7. In general, I think the author needs to strengthen the description on the implication, especially the innovation of this research.

Minor comments

Line 24: “We showed that only the outlier correction is needed to extract accurate shoreline change

25 trends.” I suggest adding a condition to this conclusion because the method is only tested for a single site.

Line 24: You mentioned only the tide correction and outlier correction are needed. Which is true?

Line 31: The author used “Shoreline positions” as different metrics here, I suggest listing what are the different shoreline positions, e.g. shoreline positions defined at different water levels.

Line 34: Please define what is high-spatiotemporal-resolution, I suggest providing specific resolutions here.

Line 79: I suggest also mentioning the development of cloud-computing platform

Line 90: If 10-15 m is medium-resolution here, what is the high-resolution defined in Line 34?

Fig1: I strongly recommend to show the MNDWI and Binary images corresponding to the true-color satellite images to visualize the impact of wet sediment and demonstrate how does it impact the CoastSat. Maybe not in Figure 1 but it’s always good to see the results visually.

Line 255~259: The author did not describe how threshold factor n is used in outlier correction, I suggest rewriting this paragraph.

Line 269: The author used “infra-gravity” here but infragravity in Line 242. I suggesting maintaining the consistency of using dash.

Line 329: I suggest mentioning the impact of the overestimation of beach slop on the tide/wave corrections.

Line 477~480: I suggest elaborating the impact of misdetection. Explain the reason why the landward tendency of SDS will result in the landward offset after applying corrections. I guess this is because the tidal-impact was over calibrated if the SDS was already on the wetland instead of instantaneous waterline. If it was this case, then the issue lays on the tidal correction instead of wave-setup correction.

---

## [Reviewer Report]

The work presented seeks to verify that the SDS obtained by means of the CoastSAT tool allow an evolutionary study of a long coastal segment such as the CRLC. It is a very complete and rigorous work that, from my point of view, is of great interest insofar as it clearly proves how the use of the SDS obtained automatically allows monitoring large segments of territory since mid-1980s, when Landsat 5 images became available, to the present day. These results, in fact, allow the authors to conclude some key considerations on how the study area has evolved -on a seasonal and decadal scale- demonstrating the real usefulness that can be derived from the systematic use of the SDS.

From my point of view this type of solutions potentially have a global applicability and this should be made more explicit in the paper. It is logical that the authors emphasize the problems that appear in those beaches with very low slope, high energy and high tidal range where the wetted area is very large and it is specially complicated to define the position of the water/land boundary since it is there where the system is being testing. However, at least in the Introduction, it should be point out that in other environments-such as low-energy microtidal beaches found in the Mediterranean, Baltic or Caribbean coasts- the applicability of the SDS series to characterize beach changes is potentially more direct.

I do not quite agree with the authors when they point out that the accuracies than can be obtained from Sentinel-2 or Landsat images are around 10m. There are examples that prove that in some other environments the accuracies are substantially better: Sánchez-García et al. (2020) evaluated the accuracy of the SHOREX system in Cala Millor, Mallorca island (Spain), on a Mediterranean sandy beach with a very low tidal range (less than 0.2 m) and low energy obtained a RMSE of 3 m for Sentinel-2 images and 3.6 m form Landsat 8 images. With this same tool, in the Portuguese Algarve beaches (meso-tidal and medium energy) Cabezas-Rabadán et al. (2020) reports RMSE values lower than 5 m using Sentinel 2 and lower than 6m with Landsat 8 images.It would be important for the authors to clarify the number of SDS they have used for their analysis, whether the number of records have change much over time and how they have solved the problem of the swaths without data form Landsat 7 images between 2003 and the present.

Regarding the results shown in Figure 5 there is a lack of further explanation of what they indicate and how they were obtained. Panels a) and b) suggest that a solution very similar to the one presented by Cabezas-Rabadán et al. (2019) has been applied, but it is not clear to me if the whole of space and time has been modeled and if the non-beach area has been omitted or not in the calculation, i.e., the latitudinal sections where we have water areas such as Grays Harbor, Willapa Bay and Columbia River. On the other hand, in panel a) it is showing the changes for each of the dates analyzed versus the first available date? If so, this should be made clearer and it should be indicated which date is taken as origin. In the same figure 5, the PDO and Niño 3.4 indices appear, the meaning of which is not evident to a reader who hasn’t studied the ENSO phenomenon in detail. I understand that a little more information should be given to the reader about their meaning. Figure 3 presents the result of applying the successive steps in the correction of a particular transect (the 020 transect) against the field measured data. Was this particular transect chosen for any reason?

In relation to the Discussion with which the SDS determine the position of the shore, might be useful for the authors to review the work of the Cabezas-Rabadán et al. (2020) in which they note that the best accuracies for the Algarve beaches with SHOREX has been achieved if the effect on the runup is only partially applied to estimate the TWL.

I consider, therefore, that this is a great article but that it would be useful if the authors correct or respond to some of the remarks that I have been point out.

I add the references that I have cited in case they may be useful to the authors.

Cabezas-Rabadán, C., Pardo-Pascual, J. E., Palomar-Vázquez, J. M., & Fernández-Sarría, A., 2019. Characterizing beach changes using high-frequency Sentinel-2 derived shorelines on the Valencian coast (Spanish Mediterranean). Science of The Total Environment, 691, 216-231. doi: 10.1016/j.scitotenv.2019.07.084.

Cabezas-Rabadán, C.; Pardo-Pascual, J.E.; Palomar-Vázquez, J.; Ferreira, Ó., and Costas, S., 2020. Satellite derived shorelines at an exposed meso-tidal beach. Journal of Coastal Research, Special Issue No. 95, pp. 1027–1031. doi: 10.2112/SI95-200.1

Sánchez-García, E., Palomar-Vázquez, J. M., Pardo-Pascual, J. E., Almonacid-Caballer, J., Cabezas-Rabadán, C., & Gómez-Pujol, L., 2020. An efficient protocol for accurate and massive shoreline definition from mid-resolution satellite imagery. Coastal Engineering, 103732. doi: 10.1016/j.coastaleng.2020.103732

---

## [Reviewer Report]

The manuscript “Monitoring interdecadal coastal change along dissipative beaches via satellite imagery at regional scale” presents a validation and application of satellite-derived shorelines (using the CoastSat toolkit) to the Columbia River Littoral Cell in Washington, USA. The study compares SDS to a long-term monitoring program and shows that with some corrections made for setup/runup and tidal effects, SDS is a viable tool for monitoring coastal change on dissipative coastlines like the CRLC. The study then makes some suggestions about how the lower-frequency changes are linked to climate cycles such as ENSO and the PDO.

I commend the authors on a very well written manuscript which I thoroughly enjoyed reading. The study is of high relevance to the journal as satellite derived shorelines are an emerging field and dissipative beaches in particular have proved challenging to measure using SDS (and Argus imagery prior). This paper provides some useful insights into the accuracies achievable as well as some techniques that can be used to enhance this.

I think that the manuscript is at a level that it could be published in Coastal Futures with only minor changes that I believe would improve the manuscript. These suggested edits are as follows:

1) I would like to see how the SDS accuracy changes in terms of its standard deviation in addition to the RMSE and R2. I find standard deviation a much more useful measure for assessing SDS accuracy as it does not include the systematic bias that might be present. This bias is not particularly important for assessing shoreline change as the shoreline width is an arbitrary measure anyway (the change is the important part). By reporting RMSE only (e.g. an RMSE of ~20m), I think it is doing the paper a disservice into reflecting the true value of the satellite shorelines (which look like they have a significantly lower standard deviation)

2) I would like to learn a little bit more about the outliers and why they occur. The supplementary figure S2 shows that outliers can be substantial and learning about what conditions they might arise under would be useful to improve the CoastSat algorithm. Do they occur under particular stages of the tide or wave conditions? Or are they random?

3) The links to climate indices such as ENSO and PDO are not really supported by appropriate evidence. I understand that this is not the scope and I don’t think the manuscript should focus too much on this, but the manuscript makes some very important statements as to how the ENSO response is different in Washington to what is reported in California. I find this hard to see in Figure 5 – could there be a more clearer figure that demonstrates this?

As I read through the manuscript, I also made some comments regarding some minor clarifications that could be addressed (see below).

Once again, thank you to the authors for a very interesting manuscript.

Mitchell Harley (Reviewer)

LINE-BY-LINE COMMENTS

Abstract Line 20: “relative reliability” is a bit vague. Not sure what reliability means here? (I don’t think a time series can be described as reliable)

Abstract 20-24: This sentence is very long and hard to follow. Can it be broken up into shorter sentences?

Line 24: should “showed” be present tense?

Line 85: Is coastal remote sensing considered an emerging field still? Argus coastal imaging is remote sensing has been around for 30 years now. I would agree that space-based remote sensing is emerging however

Line 98 and Line 114 and line 226: wave setup at the shoreline as well?

Figure 1. The text in the legend in Figure 1a looks a bit weird? As in the aspect ratio is not quite right?

Line 167: what is the mid-beach? Is it on the berm? Or the intertidal zone? This could be clarified

Line 200: There seems to be another bias/uncertainty here that is not addressed, which is the changes in optical properties of the beach associated, due to changing lighting conditions, or how the darkness around the shoreline described in the paragraph above might be modulated by the tide/waves. Figure 1 suggests this might be important

Line 244: While not essential, are their any statistics on the accuracy of the CAWCR wave hindcast for this region?

Line 287-298: The slope calculated by the CoastSat algorithm is more similar to an “upper intertidal” slope. In Vos et al., (2020) this was calculated between MSL and MHW, which is different to what you have here. How would the results look if you calculate the upper intertidal slope?

Line 312-314: As I mention above, is it possible to report on the standard deviation of these results? This would give an indication of the spread regardless of any systematic bias (which is not so important for assessing shoreline change anyway). Also, how are these statistics calculated? Are you only comparing data when both surveys and satellite measurements were taken close to the same time? Or are you interpolating the results to do the comparison?

Line 328-329: I wonder if this overestimation is due to where you have calculated the slope in the in situ data? It looks like the slope would be steeper (and hence more in line with the CoastSat slope) if you use this definition?

Line 331: Again, I would find it useful if the standard deviation could also be reported?

Line 333: I found it a bit awkward having to loop up Table S2 in the supplementary to interpret Figure 3. Can this be moved to the main manuscript?

Line 408: “Similar patterns” with regards to the ENSO needs to be elaborated – what specific patterns are observed during El Nino/La Nina phases? As I mention above, an extra figure that shows the patterns during ENSO would help

Line 420: I find reporting the RMSE ~20m is not a true reflection as a lot of this appears due to a systematic bias? These biases could be simply due to the MNDWI index detecting a different position in the swash zone and could be corrected for as well? This is what was done in Doherty et al (2022)

Line 480: OK, I see that you address this issue here, but I think calling it a misdetection is not necessarily true? I see it as a different detection (rather than mis-detection) of the shoreline due to the particular index used. This has been observed in coastal imaging (ARGUS) shoreline detection in the past (e.g. Plant et al., 2007 “The Performance of Shoreline Detection Models Applied to Video Imagery”)

Figure S6 caption: says “same figure as per Fig. S6” – do you mean Figure 6?

---

## [Editor Report]

Dear authors,

I have read your paper with great interest, along with the 3 reviews we have received. You can see that all reviewers are favorable of your work and see the potential for regional-scale assessments of our changing coastlines. As also noted by the reviewers, there are points raised where things could be clarified, and/or additional work could further improve the manuscript. I was surprised to see that including the wave components degraded your timeseries given others have said this was quite important along similar coastlines. I do wonder if in using the Stockdon equation, you also considered the simplification she also had for IG dominated coastlines, as you have in Oregon/WA? why also choose this runup correction when others have been developed at that coastline by your co-author (Ruggiero).

I look forward to seeing the revised version. 

All the best,

Kristen Splinter

Handling Senior Editor, Coastal Futures.